# Distinct miRNA Profile of Cellular and Extracellular Vesicles Released from Chicken Tracheal Cells Following Avian Influenza Virus Infection

**DOI:** 10.3390/vaccines8030438

**Published:** 2020-08-05

**Authors:** Kelsey O’Dowd, Mehdi Emam, Mohamed Reda El Khili, Amin Emad, Eveline M. Ibeagha-Awemu, Carl A. Gagnon, Neda Barjesteh

**Affiliations:** 1Research Group on Infectious Diseases in Production Animals (GREMIP), Department of Pathology and Microbiology, Faculty of Veterinary Medicine, University of Montreal, Saint-Hyacinthe, QC J2S 2M2, Canada; kelsey.odowd@umontreal.ca (K.O.); carl.a.gagnon@umontreal.ca (C.A.G.); 2Swine and Poultry Infectious Diseases Research Center (CRIPA), Department of Pathology and Microbiology, Faculty of Veterinary Medicine, University of Montreal, Saint-Hyacinthe, QC J2S 2M2, Canada; 3Department of Pathology and Microbiology, Faculty of Veterinary Medicine, University of Montreal, Saint-Hyacinthe, QC J2S 2M2, Canada; seyedmehdi.emam@mcgill.ca; 4McGill University Research Centre on Complex Traits (MRCCT), Department of Human Genetics, Faculty of Medicine, McGill University, Montreal, QC H3G 0B1, Canada; 5Department of Electrical and Computer Engineering, Faculty of Engineering, McGill University, Montreal, QC H3A 0E9, Canada; mohamed.elkhili@mail.mcgill.ca (M.R.E.K.); amin.emad@mcgill.ca (A.E.); 6Sherbrooke Research & Development Centre, Agriculture and Agri-Food Canada, Sherbrooke, QC J1M 0C8, Canada; eveline.ibeagha-awemu@canada.ca

**Keywords:** chicken, microRNA, extracellular vesicle, innate antiviral response, avian influenza virus

## Abstract

Innate responses provide the first line of defense against viral infections, including the influenza virus at mucosal surfaces. Communication and interaction between different host cells at the early stage of viral infections determine the quality and magnitude of immune responses against the invading virus. The release of membrane-encapsulated extracellular vesicles (EVs), from host cells, is defined as a refined system of cell-to-cell communication. EVs contain a diverse array of biomolecules, including microRNAs (miRNAs). We hypothesized that the activation of the tracheal cells with different stimuli impacts the cellular and EV miRNA profiles. Chicken tracheal rings were stimulated with polyI:C and LPS from *Escherichia coli* 026:B6 or infected with low pathogenic avian influenza virus H4N6. Subsequently, miRNAs were isolated from chicken tracheal cells or from EVs released from chicken tracheal cells. Differentially expressed (DE) miRNAs were identified in treated groups when compared to the control group. Our results demonstrated that there were 67 up-regulated miRNAs, 157 down-regulated miRNAs across all cellular and EV samples. In the next step, several genes or pathways targeted by DE miRNAs were predicted. Overall, this study presented a global miRNA expression profile in chicken tracheas in response to avian influenza viruses (AIV) and toll-like receptor (TLR) ligands. The results presented predicted the possible roles of some DE miRNAs in the induction of antiviral responses. The DE candidate miRNAs, including miR-146a, miR-146b, miR-205a, miR-205b and miR-449, can be investigated further for functional validation studies and to be used as novel prophylactic and therapeutic targets in tailoring or enhancing antiviral responses against AIV.

## 1. Introduction

Avian influenza viruses (AIVs) are negative-sense single-stranded RNA viruses belonging to the *Orthomyxoviridae* family. AIVs cause transmissible viral infections affecting several species of poultry, such as chickens and turkeys, as well as wildfowl. These viruses are classified into two pathotypes that are based on their pathogenesis in poultry: low pathogenic and highly pathogenic viruses. Additionally, AIVs cause respiratory infections, declines in production, and elevated mortality rates.

During the early stages of the infection, innate responses provide the first line of defense against the infections at mucosal surfaces by aiming to block the entry of the virus and viral replication. However, when the virus crosses the primary barrier, the airway epithelial cells become the target of the virus. Host cells detect the presence of viral components and, subsequently, antiviral responses will be induced. Previously, we confirmed the induction of antiviral responses in the chicken respiratory system [1,2,3]. In the absence of appropriate antiviral responses against the virus, viral replication in the host cells leads to pathological changes, such as necrosis of tracheal epithelial cells and the infiltration of inflammatory cells into the site of infection.

Furthermore, innate responses deliver information about the virulence, pathogenicity, and presence of pathogens within the host cells. In addition, immune responses influence the communication and coordination between the different host cells [4]. The induction of innate antiviral responses, the production of antiviral factors, and the regulation of viral activity are highly regulated processes that largely depend on intracellular communication mechanisms, including cell-to-cell contact, secretion of cytokines and chemokines, as well as the release of extracellular vesicles (EVs) [5]. Exosomes are membrane-encapsulated EVs measuring approximately 30 to 150 nm in diameter that can act as mediators of intracellular communication [6,7]. Studies in mammalian species have shown that EVs, including exosomes, contain a diverse array of biomolecules, including proteins, lipids, messenger RNAs (mRNAs), and microRNAs (miRNAs) [5]. They can transfer their contents to neighboring cells. In addition, there is some evidence that EVs and their contents play a role in modulating antiviral responses, but this phenomenon and the mechanism by which it occurs is poorly characterized in chickens [8,9,10,11].

MicroRNAs are conserved, small, non-coding RNA molecules that regulate the gene expression of complementary mRNA sequences, usually in the 3′ untranslated region (UTR), by base pairing, resulting in gene silencing through translational repression or target degradation [12,13]. Complete complementarity between a miRNA and its target sequence is rare in animals, but a match of as little as six base pairs in the seed region of the miRNA (nucleotides 2–8) can be sufficient to induce the suppression of gene expression [14]. As a result, a given miRNA can have multiple targets affecting multiple intracellular signaling pathways. MiRNAs have been shown to regulate several biological processes, such as development, differentiation, organogenesis, growth control, and apoptosis [15]. Extracellular miRNAs, which are packaged in exosomes, microvesicles, or apoptotic bodies, are stable and can survive in the harsh conditions of the extracellular environment. MiRNAs are also found intracellularly, but are subjected to more rapid degradation in the extracellular environment [16]. Regardless of their localization, miRNAs play an important role in cell-to-cell communication and the modulation of intracellular signaling. Furthermore, studies characterizing miRNA profiles revealed that certain miRNAs are induced upon infection with viruses and can modulate antiviral responses by targeting intracellular pathways [17,18,19,20,21]. Recent studies in humans have also demonstrated that cellular miRNAs are capable of regulating translation and replication in RNA viruses by directly binding to the virus genome [22,23,24,25,26,27]. In addition, studies in chickens have identified several miRNAs that are involved in immune responses against avian pathogens, such as avian pathogenic *Escherichia coli* (APEC), Newcastle disease virus (NDV), and Marek’s disease virus (MDV) [28,29,30,31]. Moreover, previous studies demonstrated the expression of miRNAs following AIV infections in chickens [32,33].

Despite an increasing understanding of the induction of antiviral responses in chickens, studies are required in order to determine the mechanisms by which EVs, and EV and cellular miRNAs influence and regulate antiviral responses. In this study, we aimed to determine the miRNA contents of EVs released from chicken tracheal cells following AIV infection. Moreover, we were interested in identifying whether the type of stimuli or infection can impact the profile of cellular and EV miRNAs. Therefore, in the current study, we characterized the profile of miRNAs and their potential targets that are involved in antiviral responses following the infection of chicken tracheal cells with AIV and stimulation with TLR3 and 4 ligands.

## 2. Materials and Methods 

### 2.1. Avian Influenza Virus (AIV)

Eleven-day-old specific-pathogen-free (SPF) embryonated chicken (layer chickens, white Leghorn) eggs (Canadian Food Inspection Agency, Ottawa, ON, Canada) were used to propagate influenza virus A/Duck/Czech/56 (H4N6), a low pathogenic avian influenza virus (LPAIV) by inoculation through the allantoic cavity [34]. For the inoculation, the eggs were candled, and then the virus was injected with 100 µL of stock allantoic fluid containing 0.2 hemagglutinin units (HAU) of the H4N6 virus. The allantoic fluid was harvested 48 h post-inoculation. The virus titer was determined using end-point dilution in Madin-Darby Canine Kidney (MDCK) cells (a gracious gift from Dr. Shayan Sharif’s laboratory at the Ontario Veterinary College, University of Guelph, ON, Canada) and hemagglutinin assay (HA) [35].

### 2.2. Toll-Like Receptor (TLR) Ligands

Lipopolysaccharide (LPS) from *Escherichia coli* 026:B6 (Sigma–Aldrich, Oakville, ON, Canada) and polyinosinic:polycytidylic acid (polyI:C) (InvivoGen, San Diego, CA, USA) were used in this study. These ligands were selected, as they were previously shown to induce immune responses in chicken tracheal cells [2,36].

### 2.3. Tracheal Organ Culture (TOC)

Tracheal organ culture (TOC) was performed, as previously described [2]. Briefly, tracheas were aseptically collected from 19-day-old SPF chicken embryos (Canadian Food Inspection Agency, Ottawa, ON, Canada) and, subsequently, washed twice with warm Hanks’ balanced salt solution (HBSS, Gibco, Burlington, ON, Canada) in order to remove excess mucus. The connective tissues surrounding the trachea were removed by thorough dissection. Tracheas were manually dissected into 1 mm rings using razor blades and then transferred into 24-well cell culture plates containing phenol red-free complete Medium 199 (Sigma–Aldrich, Oakville, ON, Canada). Complete Medium 199 was supplemented with 10% EV-depleted and heat-inactivated fetal bovine serum (FBS, Gibco, Burlington, ON, Canada), 25 mM 4-(2-hydroxyethyl)-1-piperazineethanesulfonic acid (HEPES) buffer (Gibco, Burlington, ON, Canada), 200 U/mL penicillin/80 µg/mL streptomycin (Gibco, Burlington, ON, Canada), and 50 µg/mL gentamicin (Gibco, Burlington, ON, Canada). To prepare EV-depleted FBS, FBS was heat-inactivated at 56 °C for 30 min. and then depleted of EVs by ultracentrifugation. To this end, complete Medium 199 containing 20% FBS were centrifuged overnight (18 h) at 100,000× *g* at 4 °C. The top layer was collected, and the resulting 20% FBS-depleted medium was then filtered through a 0.2 μm filter and diluted to the half with complete Medium 199 without FBS to reach 10% of FBS. The rings were then incubated at room temperature on a low-speed benchtop rocker for 3 h to exclude any possible reaction and mucus production during the TOC preparation. Following the incubation, the media was replaced with fresh media. During the experiments, the ciliary activity of the tracheal rings was observed under a light microscope in order to monitor the condition of TOCs and confirm the cilia activity of TOCs.

### 2.4. TOC Infection with AIV (H4N6) and Stimulation with TLR Ligands

Stimulation for all treatment groups was done in complete FBS-free Medium 199 as animal sera contain non-specific inhibitors of influenza viruses [37]. For infection of TOCs with AIV (H4N6), tracheal rings were infected with 104 pfu/mL. For the stimulation of TOCs with TLR ligands, tracheal rings were stimulated with either LPS (1 µg/mL) or polyI:C (25 µg/mL) (doses were selected based on previous studies in chickens) [2,36,38]. The control groups received complete FBS-free Medium 199. The final volume for each well was 500 µL. After 2 h of stimulation, tracheal rings were washed twice and incubated at 37 °C in fresh complete FBS-free Medium 199. The tracheal rings designated for cellular miRNA extraction (TOC samples), consisting of six biological replicates (six individual embryos) per treatment group, were collected 3- and 18-h post-infection or -stimulation and stored at −80 °C in RNAlater (Invitrogen, Burlington, ON, Canada). The tracheal supernatants of the rings that were designated for EV isolation and subsequent EV miRNA extraction (EV samples) were collected 24-h post-infection or -stimulation. For EV samples, there were three biological replicates (pools of five individual embryos per replicate) for each treatment group.

### 2.5. Extracellular Vesicle Isolation

All of the centrifugations and ultracentrifugations were performed at 4°C. Tracheal supernatants from EV samples were centrifuged at 300× *g* for 10 min. to remove cellular debris. Subsequently, supernatants were recovered and centrifuged at 2000× *g* for 20 min. Supernatants were again recovered and ultracentrifuged at 10,000× *g* for 30 min. (Optima L-100XP, Beckman Coulter, Mississauga, ON, Canada). Supernatants were recovered and filtered with 0.2 µm filters (VWR, Montreal, QC, Canada). Supernatants were then ultracentrifuged at 100,000× *g* for 60 min. (Optima L-100XP, Beckman Coulter, Mississauga, ON, Canada). Supernatants were discarded and pellets were resuspended in FBS-free complete Medium 199 and ultracentrifuged at 100,000× *g* for a final 60 min. (Optima L-100XP, Beckman Coulter, Mississauga, ON, Canada). The supernatants were discarded, and the pellets were resuspended in 200 µL of phosphate-buffered saline (PBS, Gibco, Burlington, ON, Canada). Finally, for miRNA isolation from EV samples, 700 µL of QIAzol reagent (QIAGEN, Toronto, ON, Canada) were added to samples that were designated for RNA isolation and then stored at −80 °C. Samples designated for Western Blot and transmission electron microscopy analyses were stored at −80 °C in PBS.

### 2.6. Western Blot

Western Blot analysis was performed to confirm biomarkers compatible with EVs. The protein concentrations of the isolated EVs were determined using Micro BCA Protein Assay Kit according to manufacturer’s instructions (Thermo Fisher Scientific, Burlington, ON, Canada). EV samples were lysed with RIPA Lysis Buffer (EMD Millipore, Oakville, ON, Canada) and incubated on ice for 5 min. The samples were then treated with NuPAGE LDS Sample Buffer (4X) containing 5% 2-mercaptoethanol (Invitrogen, Burlington, ON, Canada) and denatured for 5 min. at 95 °C. Equal amounts of proteins (about 30 μg) were separated by SDS-PAGE using pre-cast Mini-PROTEAN TGX electrophoresis gels (Bio-Rad, Montreal, QC, Canada). The separated proteins were transferred onto activated polyvinylidene difluoride (PVDF) membranes (Sigma–Aldrich, Oakville, ON, Canada). The membranes were washed in Tris-buffered saline with 0.05% Tween 20 (TBS-T), blocked in 5% skimmed milk TBS-T solution, and then washed again in TBS-T. The membranes were incubated with the primary antibody overnight at 4 °C, followed by the appropriate secondary antibody diluted in TBS-T for 1h at room temperature. Proteins were detected using Clarity Max enhanced chemiluminescence (ECL) Substrate (Bio-Rad, Montreal, QC, Canada), according to the manufacturer’s instructions.

The primary antibodies used for western blot were rabbit polyclonal antibody (pAb) anti-lysosome-associated membrane protein 1 (LAMP1-lysosome marker) (ab24170, Abcam, Cambridge, MA, USA), rabbit pAb anti-tumor susceptibility gene 101 (TSG101) (ab225877, Abcam, Cambridge, MA, USA), rat monoclonal antibody (mAb) anti-glucose-regulated protein 94 (GRP94) (ab2791, Abcam, Cambridge, MA, USA), and mouse mAb conjugated HRP against β-actin (HRP SC-47778, Santa Cruz Biotechnology, Dallas, TX, USA). The secondary antibodies used for antibodies were goat-anti-rabbit IgG Fc (HRP) (4041-05, Southern Biotech, Birmingham, AL, USA) and goat pAb to rat IgG (HRP) (ab97057, Abcam Cambridge, MA, USA).

### 2.7. Negative Staining and Transmission Electron Microscopy (TEM)

TEM analysis was performed at the Facility for Electron Microscopy Research (FEMR), McGill University, to confirm and validate the purity of isolated EVs. Briefly, a 1:1 dilution of frozen EV samples was thawed and fixed with 2.5% glutaraldehyde in 0.1M sodium cacodylate buffer. The samples were then allowed to equilibrate at room temperature for 30 min. A 200-mesh copper TEM grid with carbon support film (Agar Scientific Ltd., Stansted, UK) was negatively glow discharged at 25 mA for 30 s (Electron Microscopy Sciences 100 Glow Discharge System, Hatfield, PA, USA) and loaded with a 5 µL droplet of sample for 3 min. Excess solution was carefully blotted off with Whatman Grade 1 filter paper and the grid was washed twice with a droplet of glycine for 1 and 2 min., respectively. The grid was then washed three times with a droplet of MilliQ water for 1 min. Finally, for negative staining, the grid was first floated on a droplet of 2% uranyl acetate (Electron Microscopy Sciences, Hatfield, PA, USA) to remove excess water and then on a second droplet for 1 min. Excess liquid was blotted off with Whatman Grade 1 filter paper. The samples were imaged by the FEI Tecnai G2 Spirit 120 kV TEM (Thermo Fisher Scientific, Hillsboro, OR, USA) equipped with a Gatan Ultrascan 4000 CCD camera Model 895 (Gatan, Inc., Warrendale, PA, USA). The micrographs were taken at appropriate magnifications to record the fine structure of EVs. The proprietary Digital Micrograph 16-bit images (DM3) were converted to unsigned 8-bit TIFF images.

### 2.8. MiRNA Isolation

Small RNAs were isolated while using the miRNeasy Mini Kit (QIAGEN, Toronto, ON, Canada) following the Quick-Start protocol of miRNeasy Mini Kit manufacturer’s instructions. Briefly, tracheal rings from TOC samples (stored in RNAlater) were collected, lysed in 700 µL of QIAzol reagent (QIAGEN, Toronto, ON, Canada), and homogenized for two minutes using 0.5 mm glass beads (Biospec Products Inc., Bartlesville, OK, USA) and a tissue homogenizer (MP FastPrep-24 Classic Instrument, MP Biomedicals, Solon, OH, USA). The isolated EVs from EV samples were likewise lysed in 700 µL of QIAzol reagent (QIAGEN, Toronto, ON, Canada). The samples were then deproteinized in chloroform and centrifuged for 15 min. at 12,000× *g* at 4 °C. The upper aqueous portion of the samples was precipitated in 95% ethanol. The samples were sequentially washed and centrifuged with RWT and RPE buffers using RNeasy Mini columns in 2 mL collection tubes (QIAGEN, Toronto, ON, Canada). Finally, the purified RNA was eluted in 27 μL RNase-free water and quality control of RNA was performed using the RNA ScreenTape Analysis kit (Agilent Technologies, Santa Clara, CA, USA) and the Agilent 4200 TapeStation Analysis Software A.02.01 SR1 (Agilent Technologies, Santa Clara, CA, USA), according to the manufacturer’s instruction. The samples were then stored at −80 °C.

### 2.9. Small RNA Library Preparation and Sequencing

MiRNA sequencing was performed at the Research Center of the CHU de Québec-Université Laval, Quebec, Canada. Twenty-four libraries for TOC samples (pooling of two replicates from each treatment group) to give three replicates per group and twelve libraries for EV samples to have three replicates within each group were prepared using the NEBNext Multiplex Small RNA Library Prep Set for Illumina (New England Biolabs, Ipswich, MA, USA), according to manufacturer’s instructions. The total RNA from each sample was sequentially ligated to 3′ and 5′ small RNA adapters using T4 RNA ligase (New England Biolabs, Ipswich, MA, USA). Next, cDNAs were synthesized through reverse transcription using ProtoScript II Reverse Transcriptase (New England Biolabs, Ipswich, MA, USA) and amplified by PCR. Clean up and size selection of fragments was made using the Monarch PCR & DNA Cleanup Kit (5 μg) (New England Biolabs, Ipswich, MA, USA) and 6% polyacrylamide gel. Finally, the RNA libraries were sequenced (50 bp single-end) on an Illumina HiSeq 2500 platform (rapid mode) (Illumina, San Diego, CA, USA).

### 2.10. MiRNA Expression Analysis

Raw sequencing reads were processed by removing adapters and low-quality sequences using Trimmomatic, a read trimming tool for Illumina NGS data [39]. We used FastQC, a quality control tool for high throughput sequencing data, and MultiQC v1.7, a tool that creates a single report for different metrics and alignment statistics across many samples, in order to assess the quality of the generated reads [40,41]. We aligned the clean reads obtained from each library to mature miRNAs and to the reference genome of *Gallus gallus* in miRBase version 22.1 database to identify known miRNA expression levels in each group [42]. The MirDeep2 package, which maps reads against a library of known miRNAs from miRBase, was used for miRNA quantification from the reads, coordinated by a Nextflow workflow [43,44]. Small RNA sequencing libraries were normalized to counts per million (CPM) and evaluated for expression. MiRNA differential expression modeling and calculation were then done using R and R packages *edgeR*, *tidyverse*, *magrittr* and *ComplexHeatmap* [45,46,47,48,49]. Furthermore, the identified miRNAs for each treatment group were considered differentially expressed (DE) if their normalized expression fold changes (FC) relative to the control group was greater than or equal to 1.5-fold (log2FC ≥ 0.58 or log2FC ≤ −0.58) and if their False Discovery Rate (FDR) was less than 0.05 (FDR < 0.05). Venn diagram analysis for the DE miRNAs among the different treatment groups was performed using the online tool http://bioinformatics.pbs.ugent.be/webtools/Venn/. Furthermore, the distribution and intersection of the DE miRNAs were visualized using the UpSet software (http://vcglab.org/upset) [50].

### 2.11. In Silico Target Gene Prediction and Pathway Analysis

We downloaded targets of *Gallus gallus*’ miRNAs from release 7.2 of TargetScan and version 6.0 of miRDB in order to perform functional enrichment analysis on the targets of DE miRNAs [51,52]. Next, we excluded targets with a low confidence score (below 99 ‘context++ score percentile’ for TargetScan and below 95 ‘target prediction score’ for miRDB), resulting in 4987 unique target genes for 982 miRNAs from TargetScan and 4887 unique target genes for 973 miRNAs from miRDB. Subsequently, we formed gene sets corresponding to targets of differentially expressed, up-regulated, and down-regulated miRNAs, according to TargetScan and miRDB (separately). We excluded genes that were targets of both up-regulated and down-regulated miRNAs in order to calculate the exact number of host target genes or host target pathways.

We used the KnowEnG analytical platform (https://knoweng.org/analyze) to perform pathway and gene ontology (GO) enrichment analysis on these gene sets [53,54]. KnowEnG is a computational system for analysis of ‘omic’ datasets in light of prior knowledge in the form of various biological networks. We used the KnowEnG’s gene set characterization (GSC) pipeline in the standard mode (no knowledge network) and its knowledge-guided mode with the STRING co-expression network [55]. The knowledge-guided mode of this pipeline implements DRaWR, a method that utilizes random walk with restarts (RWR) in order to incorporate gene-level biological networks in the enrichment analysis to improve identification of important pathways and GO terms [56]. For the GSC pipeline, we did not use the bootstrapping option, selected *Gallus gallus* as ‘species’, and used default values for all other parameters. In addition, we mapped the *Gallus gallus* genes to *Homo sapiens* genes and used them in the same manner in the GSC pipeline with ‘species’ selected as *Homo sapiens*. The p-values of the Fisher’s exact test corresponding to the standard mode of GSC pipeline were corrected for multiple hypotheses (Benjamini–Hochberg method) and FDR < 0.05 was considered to be significant. For the network-guided mode of GSC pipeline, the results were filtered based on the ‘Difference Score’ and those with a value larger than 0.5 were considered. The difference score is the normalized difference between the query probabilities and the baseline probabilities in the RWR algorithm, with the best score observed as one [56]. Finally, we used GraphPad Prism 8.4.3 for the illustration of the targeted pathways by the treatment group [57].

In addition, we performed the prediction of candidate miRNAs targeting influenza viral genes. The genomic sequences for the AIV strain used in this experiment, A/Duck/Czech/56 (H4N6) (genome accessions: CY130022, CY130023, CY130024, CY130025, CY130026, CY130027, CY130028, CY130029), were obtained from the Influenza Virus Resource at the National Center for Biotechnology Information [58,59]. Two platforms, miRanda and RNAhybrid, were used to scan the eight segments of the AIV viral genome for potential target sites of the identified DE miRNA. The source code for miRanda, (written in C and downloaded from http://www.microrna.org/microrna/getDownloads.do) was used with default parameters for scaling parameter (4.0), strict 5′ seed pairing (off), gap-opening penalty (−4.0), and gap-extend penalty (−9.0), and adjusted parameters for score, set to greater than or equal to 160 (sc ≥ 160), and the minimum free energy (mfe), set to less than or equal to −16 kcal/mol (en ≤ −16 kcal/mol), as previously suggested [60,61]. Identified miRNA target pairs were confirmed using RNA hybrid (http://bibiserv.techfak.uni-bielefeld.de/rnahybrid/submission.html), which illustrates hybridization based on mfe [62].

## 3. Results

### 3.1. Chicken Tracheal Cells Release Extracellular Vesicles

While there is heterogeneity among EV cargo, depending on cell type and origin, the protein contents of extracellular vesicles and exosomes have been extensively studied [6].

The average protein concentration across all EV samples was 506 μg/mL. Based on the International Society for Extracellular Vesicles (ISEV)’ recommendation, our results highlighted the presence of “EV-enriched” markers and the absence of non-exosomal markers in EV samples by Western Blot analysis [63]. Our results confirmed the presence of lysosomal LAMP1, a type I transmembrane protein, in both EV and cell lysate samples (Figure 1a). The LAMP1 protein bands identified in both samples were of slightly different sizes. This was expected, as the antibody detects a band of approximately 90–130 kDa. The variability in molecular weight is observed as a result of different levels of glycosylation of the target in different cell and tissue types [64]. In addition, endosomal TSG101, a cytosolic protein, and cytoskeletal β-actin proteins were detected in EV samples (Figure 1a). Moreover, as expected, the EV samples lacked endoplasmic reticulum protein, GRP94 (Figure 1a) [63]. Finally, EVs were examined for morphological characteristics by transmission electron microscopy (TEM), which revealed vesicles of 50–150 nm in diameter with morphological characteristics of EVs (Figure 1b).

### 3.2. Cellular and EV Treatment Groups Have Distinct miRNAs Expression Profiles

RNA quality control of the extracted RNA determined an average RNA integrity number (RIN) of 6.0 and an average concentration of 406 ng/μL. Following small-RNA sequencing, low-quality reads were filtered and adaptor sequences were trimmed. A total of 235,810,088 clean reads were obtained from TOC samples (three libraries per treatment group). A total of 27,298,247 clean reads were obtained From EV samples (three libraries per treatment group) (Appendix A). Across all samples, the distribution of the small RNA sequencing length was consistent with the known length of mature miRNA, an average of 22 nucleotides [65]. The fragment lengths were primarily concentrated at 22 nucleotides (Appendix A).

The miRNA expression profiles were evaluated in order to determine the ability of AIV infection, and LPS and polyI:C stimulation to influence the expression of cellular and EV miRNAs. A total of 692 known mature chicken miRNAs were detected. Following differential expression filtering (FC ≥ 1.5 and FDR < 0.05) and mean difference plot analysis, 228 DE unique miRNAs were identified (Appendix A). In addition, some DE miRNAs were present in several treatment groups (Figure 2, Appendix A).

Within the TOC 3 h groups, a total of 18 unique DE miRNAs, of which two miRNAs were found in more than one group (Figure 3a–c). Our results demonstrated that 11, three, and two miRNAs were up-regulated, while four miRNAs were down-regulated in the TOC 3 h, AIV group (Table 1 and Table 2). There were two miRNAs, gga-miR-1608 and gga-miR-6705-5p, which were DE in both the TOC 3 h AIV and TOC 3 h LPS groups (Figure 2a, Appendix A). Furthermore, within the TOC 18 h groups, our results showed a total of 88 unique DE miRNAs, of which six miRNAs were found in more than one group (Figure 4a–c). In the TOC 18 h groups, three, 32, and two miRNAs were up-regulated, and two, 15, and 40 miRNAs were down-regulated following treatment with AIV, LPS and polyI:C, respectively (Table 1 and Table 2). There were three miRNAs, gga-miR-1451-5p, gga-miR-1563, and gga-miR-12234-5p, which were DE in both TOC 18 h AIV and TOC 18 h LPS groups and three miRNAs, gga-miR-6569-5p, gga-miR-12228-3p, and gga-miR-2184a-3p, which were DE in both the TOC 18 h LPS and TOC 18 polyI:C groups (Figure 2b, Appendix A).

With respect to the EV groups, a total of 145 unique DE miRNAs were identified, in which 59 miRNAs were found in more than one group (Figure 5a–c). Our results exhibited that 21, five, and 14 miRNAs were up-regulated in AIV, LPS, and polyI:C groups, respectively, while 57, 17, and 90 miRNAs were down-regulated in AIV, LPS, and polyI:C groups, respectively (Table 3 and Table 4). There were 10 miRNAs, gga-miR-107-5p, gga-miR-1784b-5p, gga-miR-449b-5p, gga-miR-205b, gga-miR-210a-5p, gga-miR-1727, gga-miR-1464, gga-miR-6665-5p, gga-miR-7482-5p, and gga-miR-12284-3p, which were DE in all EV groups. In addition, there was one miRNA, gga-miR-383-5p, which was DE in both the EV AIV and EV LPS groups. Furthermore, 34 miRNAs were DE in both EV AIV and EV polyI:C groups. Moreover, four miRNAs, gga-miR-211, gga-miR-132b-5p, gga-miR-1597-5p, and gga-miR-12223-3p were DE in both EV LPS and EV polyI:C groups (Figure 2c, Appendix A).

Across all TOC 3 h, TOC 18 h, and EV groups, there were 67 up-regulated miRNAs, 157 down-regulated miRNAs, and four miRNAs, gga-miR12244-5p, gga-miR210a-5p, gga-miR30b-5p, and gga-miR-383-5p, which were found to be up-regulated in some groups and down-regulated in others. The miRNA gga-miR12244-5p was up-regulated in the TOC 18 h AIV group and down-regulated in the EV AIV group and gga-miR210a-5p was up-regulated in the TOC 3 h AIV and TOC 18 h LPS groups and down-regulated in all three EV groups. The miRNA gga-miR30b-5p was up-regulated in the TOC 18 h LPS group and down-regulated in the EV polyI:C group and, finally, gga-miR-383-5p was up-regulated in the EV AIV and EV polyI:C groups and down-regulated in the TOC 3 h AIV group (Table 1, Table 2, Table 3 and Table 4).

### 3.3. Target Gene Prediction and Functional Annotation Reveals DE miRNAs Target Multiple Pathways

The miRDB and TargetScan databases were used to predict the possible gene set or pathway targets of DE miRNAs to characterize molecular and immunoregulatory functions of DE miRNAs presented in the current study. Target genes were predicted based on *Homo sapiens* (human), *Mus musculus* (mouse), and *Gallus gallus* (chicken) targets. However, there was limited information on *Gallus gallus* targets; therefore, the results based on the chicken database are provided in the Appendix A). The results presented in this study are based on human and mice databases. A total of 105, 34, and 151 projected genes and three, four, and 14 projected pathways were targeted by 15, three, and two DE cellular miRNAs in 3 h TOC AIV, LPS, and polyI:C groups, respectively. Moreover, 25, 578, and 384 projected genes and 17, 14, and 17 projected pathways were targeted by five, 47, and 42 DE miRNAs in 18 h TOC AIV, LPS, and polyI:C treatment groups, respectively (Table 5, Table 6 and Table 7). Finally, a total of 487, 147, and 817 genes and 16, 11, and 23 pathways were identified as targets for 78, 22, and 104 DE EV miRNAs following AIV, LPS, and polyI:C treatments (EV samples), respectively (Table 5, Table 8 and Table 9). The targeted pathways related to immune responses and intracellular signaling are illustrated in Figure 6, while a complete list of targeted pathways and the respectively responsible miRNAs are provided in Table 6, Table 7, Table 8 and Table 9.

Our data demonstrated that DE miRNAs may target several genes and pathways that play roles in cell physiology, cell cycle, and immune responses. Based on the target pathway predictions obtained for the TOC 3 h groups, some miRNA may regulate several pathways, such as the brain-derived neurotrophic factor (BDNF) and PI3K-Akt-mTOR-signaling pathways. In addition, the PI3K-Akt-mTOR-signaling pathway was predicted for target genes of both the up-regulated TOC 3 h LPS and up-regulated TOC 3 h polyI:C groups, and they could be regulated by different miRNAs, such as gga-miR-6704-5p and gga-miR-12235-5p, respectively (Figure 6a, Table 6).

For TOC 18 h groups, the TNF-alpha NF-κB signaling pathway was found to be targeted by gga-miR-1793, gga-miR-7457-5p (AIV group), and gga-miR-6641-5p (LPS group) and gga-miR-7480-5p (polyI:C group). Finally, for the TOC 18 h AIV treatment group, the regulation of TLR signaling pathway was found to be targeted by gga-miR-12244-5p, which was DE following AIV infection (Figure 6b,c, Table 6 and Table 7).

The results presented here demonstrated that EV up-regulated miRNAs following AIV infection potentially could target gene sets related to mRNA processing and c-myc, while other gene sets or pathways could be targeted by EV down-regulated miRNA following AIV infection, such as the TGF-beta pathway and caspase cascade in apoptosis. Some gene sets, such as c-myc or TGF-beta signaling related genes related genes, can be regulated by both up-regulated or down-regulated EV miRNAs following AIV infection.

Our results demonstrated that certain pathways were uniquely targeted by DE miRNAs in a specific treatment group. For example, it was predicted that the p73 transcription factor network may be targeted by up-regulated miRNAs following AIV infection in EV samples. Our results predicted that the caspase cascade during the apoptosis process can be regulated by gga-miR-1784b-5p, which was down-regulated in both the EV LPS and EV polyI:C groups. DE EV miRNAs following polyI:C treatment, such as gga-miR-12235-5p, gga-miR-1632-5p, gga-miR-218-5p, and gga-miR-7482-5p, are potentially able to regulate circadian rhythm related genes, direct p53 effectors and PI3K-Akt-mTOR-signaling pathway (Figure 6d,e, Table 8 and Table 9).

It was projected that some DE miRNAs, such as gga-miR-449b-5p or gga-miR-205b may target several pathways. For example, gga-miRNA-205b can potentially regulate mRNA processing, p53 transcription factor and Ras-related C3 botulinum toxin substrate 1 (RAC1) pathway (Table 9). Furthermore, gga-miR-12284-3p was also a miRNA down-regulated in all EV groups. Our data from the target gene set prediction within all three different treatments showed that this miRNA could regulate p53 transcription factor (Table 9). The gga-miR-383-5p, which was down-regulated in the TOC 3 h treatment group, was predicted to target and potentially regulate the c-myc pathway (Table 8). In addition, this miRNA was up-regulated following infection with AIV and stimulation with LPS in the EV treatment groups and it may also target the c-myc pathway based on the EV data (Table 9).

### 3.4. The Functional Annotation Reveals DE miRNAs Target Multiple Segments of the AIV Viral Genome

We also investigated the target of miRNAs within the AIV viral genome in order to determine the possible roles of DE miRNAs in AIV infection. A total of 26 miRNAs were determined to target at least one segment of the viral genome (Table 10). One miRNA, gga-miR-1784b-5p, was found to potentially target two viral segments, segment 5 (NP protein) and segment 7 (M1 protein). Figure 7 illustrates the target sites within the viral segments. Our results demonstrated that some down-regulated EV miRNAs such as gga-miR-107-5p, gga-miR-6665-5p and gga-miR-1784b-5p, possibly can target AIV segments (Table 10). Among the DE miRNAs targeting the AIV viral genome in both the TOC and EV treatment groups, 4 miRNAs were up-regulated, and 22 miRNAs were down-regulated. In addition, the secondary structures for the miRNA-RNA interactions were predicted using RNAhybrid, which was also used to predict and confirm the mfe values predicted by miRanda (Appendix A).

## 4. Discussion

Studies investigating cellular miRNAs following viral infections are essential for providing insight into the role of miRNAs in intracellular communication and the induction of antiviral responses. Identifying the mechanisms that can be regulated by miRNAs following infections or treatments will expand the current knowledge of host-pathogen interactions. This regulation is complex and, while host miRNAs can positively regulate antiviral responses, viruses have also been shown to impact miRNA expression to favor viral infection [66]. Here, we report for the first time that chicken tracheal cells secrete EVs. This work revealed that EV miRNAs have the potential to regulate antiviral responses in chickens. Furthermore, we observed that the miRNA profiles of chicken tracheal cells depend on the source (cellular versus EV) and the treatment (AIV, LPS, or polyI:C). These differences in the profile of miRNAs can be associated with active machinery of cargo loading during EV maturation and release. Previous studies have demonstrated that multiple mechanisms, including the endosomal sorting complex required for transport (ESCRT)-dependent and ESCRT-independent pathways, regulate the content of the EVs [67,68]. In addition, viral infections or stimulation via TLRs can change the EV cargo composition by targeting these pathways [8].

Additionally, we found a group of miRNAs that are in common among the different treatment groups in EV samples, while they are significantly affected by treatments. This suggests that, while the treatment groups have distinct miRNA expression profiles, certain miRNAs may have a fundamental role in which their loading into the EVs is independent of treatment or independent of the active cargo loading process.

EVs and their contents, including miRNAs, play critical roles in the regulation of the immune responses in the recipient cells by either activating or suppressing immune response genes [69,70]. In the context of influenza virus infections, previous studies demonstrated that miRNAs regulate antiviral responses by suppressing intracellular signaling pathways or activating pathways downstream of pattern recognition receptors (PRRs). For example, miR-92-5p, which was up-regulated in EVs following AIV infection, is able to enhance the activity of the NF-κB pathway [71]. Meanwhile, the expression of miR-449b-5p was suppressed following virus infection. The miR-449 family mainly interfere with influenza virus infection by enhancing type I IFNs [18]. Therefore, as highlighted in this study, the type of stimuli affected the EV cargo and, subsequently, can have potential impact on the recipient cells. It is likely to expect that the treatment of host cells with specific miRNAs in EVs that activate antiviral responses in the recipient cells may limit the replication of the virus in the neighboring cells. On the other hand, the virus may down-regulate the expression of some EV miRNAs and interfere with antiviral responses.

Network-guided gene set characterization analysis using KnowEnG’s analytical platform enabled the identification of important GO terms and pathways, while incorporating co-expression relationships among targets of DE miRNAs [53]. These pathway analyses demonstrated that DE EV miRNAs can target several pathways that are critical during influenza virus infection, such as RAC1 and TGF-beta signaling pathways. The RAC1 pathway has been shown to enhance the replication of the influenza virus [72]. Therefore, targeting this pathway, through specific miRNAs, such as miR-205a, can be a strategy to limit the replication of the influenza virus either in the target cells or neighboring cells through EVs.

Analysis of the miRNA profile in chicken tracheal cells at different time points showed that the type of stimuli impacts the profile of DE miRNAs. For instance, the majority of the differentially expressed miRNAs were down-regulated in the polyI:C treatment group, while the majority of the differentially expressed miRNAs were up-regulated in AIV infection group at an early time point. Previous studies in chickens highlighted the possible role of miRNAs in the regulation of immune responses and the modulation of Marek’s disease virus or avian influenza virus. As highlighted in the study by Wang et al., the profile of expressed miRNAs or miRNA-related mechanisms that are involved in the regulation host immune response could vary depending on the type of virus, cell stimuli, and the time point [33,73].

Moreover, as previously suggested, our analyses showed that multiple miRNAs can target the same gene and that, equally, a single miRNA can have multiple gene targets [74]. This suggests that it is potentially a combination of miRNA activities that modulate target gene expression. Among the cellular TOC samples, the distinct miRNA expression characteristics at different post-infection or -stimulation time points (3 h and 18 h) suggests that the expression of certain miRNAs is time-dependent. There were not any common DE miRNAs within each treatment group among different time points.

While the functions of most of the DE miRNAs identified in the various treatment groups remain unknown, some have been reported to be involved in the regulation of host immune responses and host-pathogen interactions. Previously, we showed that, in the same tissues, LPS is a potent TLR ligand for inducing pro-inflammatory cytokines, such as IL-6 and IL-1β cells, which limit the replication of AIV [1,2,36]. The current study showed that LPS up-regulated the expression of miR-146a in tracheal cells at a later time point. A previous study demonstrated that the activation of NF-κB signaling pathway leads to the up-regulation of miR-146a, which regulates and controls pro-inflammatory responses [75]. Moreover, miR-181 family, including miR-181a and -b, regulates pro-inflammatory responses by regulating NF-κB signaling pathway [76,77,78]. In this study, the expression of both miRNA-181a and -b was increased following LPS treatment. This result is in alignment with previous reports indicating increased expression of miR181b following LPS treatment in chicken macrophages [79]. Therefore, expressed miRNAs, such as miR-181 family or 146a following LPS treatment, can control pro-inflammatory responses through the NF-κB signaling pathway.

In addition, the results of the current study demonstrated that some miRNAs have the potential to target both innate and adaptive immune responses. For example, some DE miRNAs, such as gga-miR-205b and gga-miR-124a-5p, target p53 signaling pathway (Figure 6c,e, Tables 7,9). It has been demonstrated that p53 serves as a host antiviral factor by enhancing cytokine and antiviral gene responses in the lung, increasing the activity of dendritic cells (DC) and influenza-specific CD8+ T cells [80]. Therefore, these miRNAs could be employed to develop anti-influenza strategies and vaccine adjuvants for the control of AIV in chickens.

In addition, previous studies have also implied that miR-146 is involved in the regulation of immune responses to *Salmonella* infection in mice and it has direct targets within TLR signaling [81,82,83]. Our results indicated that gga-miR-12244-5p may target TLR signaling pathways. The miRNA gga-miR-146b-5p was previously suggested to be involved in the general process of inflammation during Avian pathogenic *Escherichia coli* (APEC) infection in chickens [28].

The DE miRNAs play a role in the regulation of antiviral responses, but they also might target the AIV genome. We were interested in miRNA target sites in the AIV genome, as recent studies have indicated that miRNAs may be capable of regulating virus translation and replication by directly binding to the virus genome [22,23,24,25,26,27]. While investigating the potential targets in the AIV viral genome, we determined that seven out of eight segments were targeted by at least one miRNA, suggesting that it is possibly a combination or cooperation of miRNA activity that can control viral activity. Our results demonstrated that both gga-miR-146a-5p and gga-miR-146b-5p, which are part of the same gene family, target segment 1 (PB2 protein) of the AIV genome. This suggests that, while gga-miR-146a-5p and gga-miR-146b-5p may play a role in modulating TLR signaling, they potentially have a dual function in regulating the PB1 protein. In addition, gga-miR-1784b-5p was the only miRNA identified that targeted more than one viral segment of AIV, both segment 5 (NP protein) and segment 7 (M1 protein). Our results suggest that this miRNA has multiple targets, both in the host and viral genomes. Several other miRNAs, gga-miR-129-5p, gga-miR-6641-5p, and gga-miR-1663-5p, target host mRNA processing and NF-kB signaling pathways, while also potentially targeting segments 2 (PB1 protein) and 3 (PA protein) of AIV.

## 5. Conclusions

In summary, we investigated the global miRNA expression profile in chicken tracheas in response to AIV infection and TLR ligand stimulation and the potential roles of specific DE miRNAs. Our results highlighted the possible role of EV contents in regulating antiviral responses. This accumulated evidence presented here elucidated the potential function of the candidate miRNAs on the host responses to AIV infection. We inferred that miRNAs play a major role in AIV infection and they can coordinate host defense against viral infections. In this study, the miRNA expression profiles were significantly regulated by the treatments with AIV, LPS, and polyI:C. These DE miRNAs were found to potentially target both host and viral genes. This study identified several miRNAs, such as miR-146a, miR-146b, miR-205a, miR205b, and miR-449, which could be employed in alternative strategies for the control of AIV in chickens, such as miRNA-based antiviral agents or vaccine adjuvants. Further functional studies are required to confirm the effect of treatment with candidate miRNA to inhibit AIV infection and induce stronger immune responses in in vitro and in vivo models.

## Figures and Tables

**Figure 1 vaccines-08-00438-f001:**
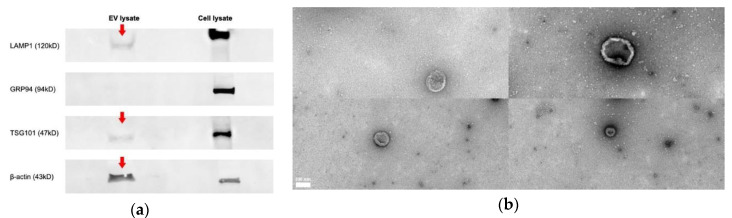
Purification and characterization of extracellular vehicles (EVs) released from tracheal organ culture (TOC). (**a**) Western Blot analysis for common exosomal marker proteins including lysosome-associated membrane protein 1 (LAMP1), tumor susceptibility gene 101 (TSG1) in EV, and cell lysate samples. Equal amounts of an isolated EV protein sample, along with cell lysate controls, were fractionated on electrophoresis gels and electro-transferred to onto activated polyvinylidene difluoride (PVDF) membranes. Detection by chemiluminescence revealed the presence of lysosomal LAMP1, endosomal TSG101, and cytoskeletal β-actin proteins and the absence of endoplasmic GRP94 in EV samples. The LAMP1 protein bands identified in both samples were of slightly different sizes. This was expected, as the antibody detects a band of approximately 90–130 kDa. The variability in molecular weight is observed as a result of different levels of glycosylation of the target in different cell and tissue types [64]. (**b**) Transmission electron microscopic images of EVs isolated and purified from the culture supernatant of TOCs. EV morphology is observed by negative staining. EVs showed their donut-shaped morphology. Scale bar = 100 nm.

**Figure 2 vaccines-08-00438-f002:**
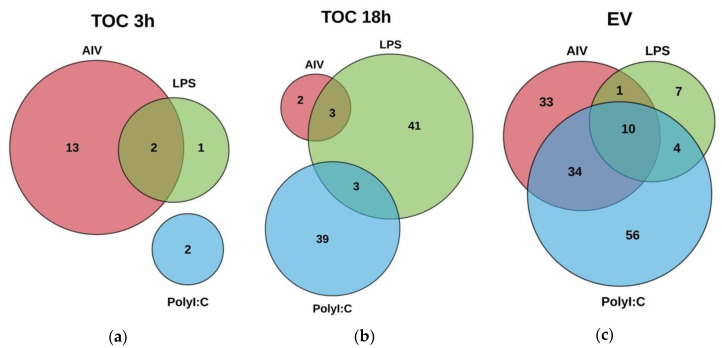
Venn diagram showing DE miRNAs in (**a**) TOC 3 h groups treated with AIV, LPS and polyI:C, (**b**) TOC 18 h groups treated with AIV, LPS and polyI:C and (**c**) EV groups treated with AIV, LPS and polyI:C. Lists of DE miRNAs for TOC 3 h, TOC 18 h, and EV treatment groups are shown in Appendix A, respectively.

**Figure 3 vaccines-08-00438-f003:**
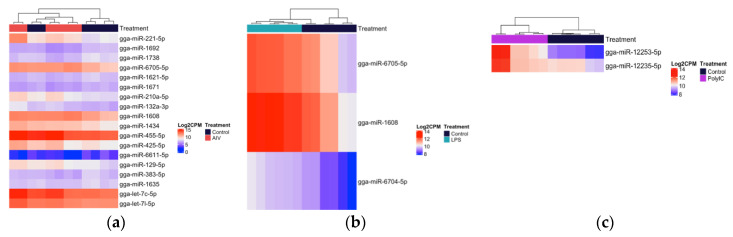
Heatmap with hierarchical clustering of miRNA distribution of the top cellular differentially expressed (DE) miRNAs (3 h) ranked by adjusted p-value (FDR). (**a**) The top 18 DE miRNAs (of 18 that pass the FDR threshold), AIV group. (**b**) The top three DE miRNAs (of three that pass the FDR threshold), LPS group. (**c**) The top two DE miRNAs (of two that pass the FDR threshold), polyI:C group. FDR < 0.05. The density of the colors represents the abundance of each miRNA in the scale of log2 counts per million (CPM).

**Figure 4 vaccines-08-00438-f004:**
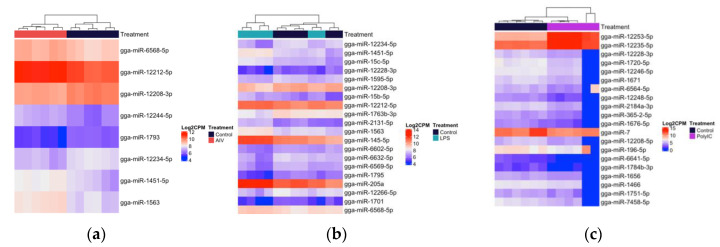
Heatmap with hierarchical clustering of miRNA distribution of the top cellular DE miRNAs (18 h) ranked by adjusted p-value (FDR). (**a**) The top eight DE miRNAs (of eight that pass the FDR threshold) AIV group. (**b**) The top 20 DE miRNAs (of 65 that pass the FDR threshold), LPS group. (**c**) The top 20 DE miRNAs (of 47 that pass the FDR threshold) polyI:C group, FDR < 0.05. The density of the colors represents the abundance of each miRNA in the scale of log2 counts per million (CPM).

**Figure 5 vaccines-08-00438-f005:**
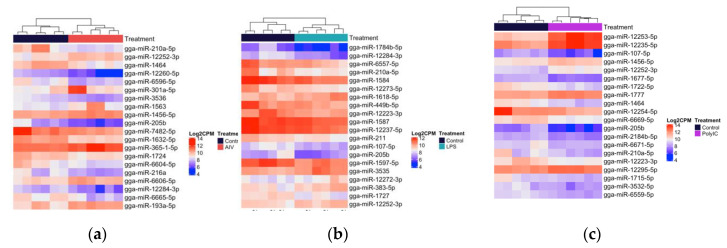
Heatmap with hierarchical clustering of miRNA distribution of the top EV DE miRNAs ranked by adjusted *p*-value (FDR). (**a**) The top 20 DE miRNAs (of 99 that pass the FDR threshold) AIV group. (**b**)The top 20 DE miRNAs (of 31 that pass the FDR threshold), LPS group. (**c**) The top 20 DE miRNAs (of 147 that pass the FDR threshold) polyI:C group. FDR < 0.05. The density of the colors represents abundance of each miRNA in the scale of log2 counts per million (CPM).

**Figure 6 vaccines-08-00438-f006:**
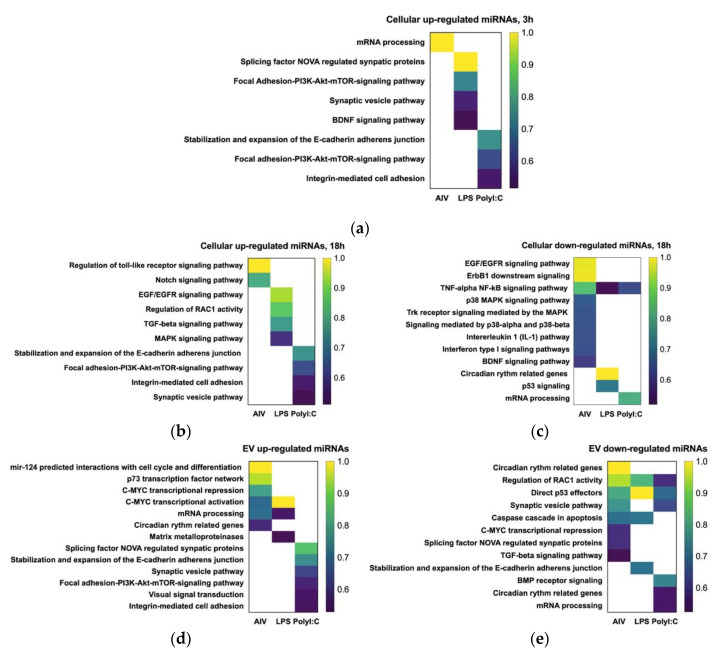
Pathway and gene ontology (GO) enrichment analysis on target genes of (**a**) up-regulated cellular (3 h), (**b**) up-regulated cellular (18 h), (**c**) down-regulated cellular (18 h), (**d**) up-regulated EV, and (**e**) down-regulated EV DE miRNA following treatment with AIV, LPS and PolyI:C. KnowEnG analytical platform (https://knoweng.org/analyze) was used to perform pathway and gene ontology (GO) enrichment analysis on these gene sets based on human database. The GO category analysis based on biological process for targets of DE miRNAs and the color intensities indicate the difference scores obtained from pathway analysis. Down-regulated (3 h) is not shown as there were no down-regulated DE miRNAs for both the TOC 3 h LPS and TOC 3 h polyI:C groups. The density of the colors represents the difference score. FDR < 0.05 and difference score >0.5.

**Figure 7 vaccines-08-00438-f007:**
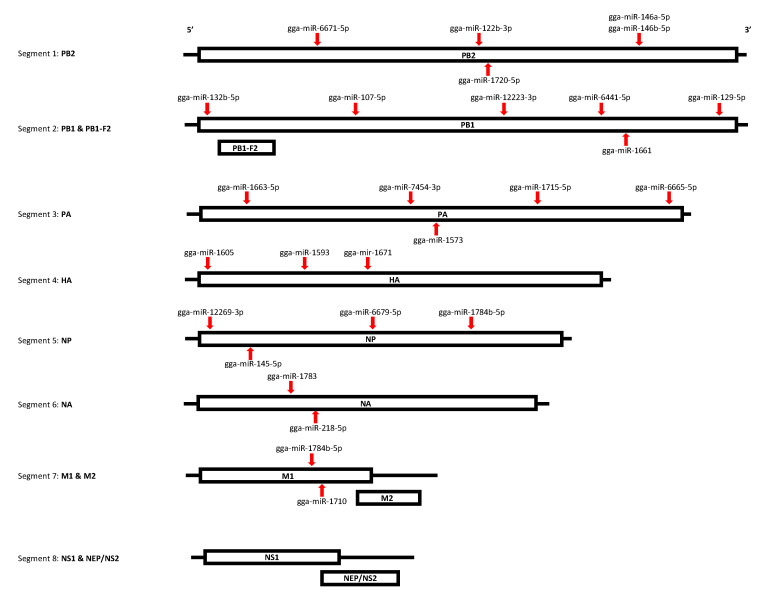
MiRNA target sites within AIV viral genome. The target sites for miRNAs within the AIV viral genome were predicted using the miRanda and RNAhybrid algorithms. All of the segments were found to be targeted by at least one miRNA, except segment 8 (NS1, NEP). One miRNA, gga-miR-1784b-5p, was found to target both segments 5 (NP protein) and 7 (M1 protein). Specific target positions for each miRNA can be found in Table 10.

**Table 1 vaccines-08-00438-t001:** Up-regulated miRNAs in TOC 3 h and TOC 18 h following treatment with avian influenza viruses (AIV), lipopolysaccharide (LPS), and polyI:C. Following differential expression filtering (FC ≥ 1.5 and FDR < 0.05), 11, three, and two miRNAs were found to be up-regulated in TOC 3 h AIV, LPS, and polyI:C groups, respectively. For TOC 18 h AIV, LPS, and polyI:C groups, three, 32, and two miRNAs were found to be up-regulated in each group, respectively.

Treatment Group		MiRNA	Log_2_ Fold Change	Fold Change
TOC 3 h	TOC AIV 3 h (11 miRNAs)	gga-miR-221-5p	2.23	4.690
		gga-miR-425-5p	0.991	1.987
		gga-miR-210a-5p	0.956	1.940
		gga-miR-455-5p	0.904	1.871
		gga-miR-6705-5p	0.845	1.796
		gga-miR-1608	0.806	1.749
		gga-let-7c-5p	0.786	1.724
		gga-miR-132a-3p	0.767	1.701
		gga-miR-1434	0.751	1.683
		gga-miR-129-5p	0.701	1.625
		gga-let-7l-5p	0.629	1.547
	TOC LPS 3 h (3 miRNAs)	gga-miR-6705-5p	1.087	2.124
		gga-miR-1608	1.053	2.075
		gga-miR-6704-5p	1.023	2.032
	TOC polyI:C 3 h (2 miRNAs)	gga-miR-12253-5p	2.716	6.568
		gga-miR-12235-5p	1.055	2.078
TOC 18 h	TOC AIV 18 h (3 miRNAs)	gga-miR-1563	0.742	1.673
		gga-miR-1451-5p	0.722	1.649
		gga-miR-12244-5p	0.702	1.627
	TOC LPS 18 h (32 miRNAs)	gga-miR-1451-5p	0.97	1.959
		gga-miR-425-5p	0.948	1.929
		gga-miR-221-5p	0.887	1.849
		gga-miR-191-5p	0.882	1.843
		gga-miR-145-5p	0.858	1.813
		gga-miR-30c-5p	0.851	1.804
		gga-miR-146b-5p	0.851	1.803
		gga-miR-15b-5p	0.81	1.753
		gga-miR-30b-5p	0.808	1.750
		gga-miR-15c-5p	0.801	1.742
		gga-miR-210a-5p	0.795	1.735
		gga-miR-2131-5p	0.794	1.734
		gga-miR-146a-5p	0.787	1.725
		gga-miR-2184a-5p	0.781	1.719
		gga-miR-1563	0.777	1.713
		gga-miR-455-5p	0.769	1.704
		gga-miR-205a	0.767	1.702
		gga-let-7b	0.76	1.694
		gga-miR-181b-5p	0.756	1.689
		gga-miR-30a-5p	0.753	1.685
		gga-miR-30d	0.753	1.685
		gga-miR-1729-5p	0.752	1.685
		gga-miR-16c-5p	0.725	1.653
		gga-miR-17-5p	0.725	1.653
		gga-let-7c-5p	0.708	1.634
		gga-miR-10c-5p	0.708	1.633
		gga-miR-181a-5p	0.691	1.614
		gga-miR-18a-5p	0.685	1.608
		gga-miR-10a-5p	0.671	1.592
		gga-miR-16-5p	0.651	1.570
		gga-let-7l-5p	0.63	1.547
		gga-miR-1559-5p	0.613	1.529
	TOC polyI:C 18 h (2 miRNAs)	gga-miR-12253-5p	3.565	11.831
		gga-miR-12235-5p	1.503	2.834

**Table 2 vaccines-08-00438-t002:** Down-regulated miRNAs in TOC 3 h and TOC 18 h following treatment with AIV, LPS and polyI:C. Following differential expression filtering (FC ≥ 1.5 and FDR < 0.05), four, zero, and zero miRNAs were found to be down-regulated in TOC 3 h AIV, LPS, and polyI:C groups, respectively. For TOC 18 h AIV, LPS, and polyI:C groups, 2, 15, and 40 miRNAs were found to be down-regulated in each group, respectively.

Treatment Group		MiRNA	Log_2_ Fold Change	Fold Change
TOC 3 h	TOC AIV 3 h (4 miRNAs)	gga-miR-383-5p	−0.591	−1.507
		gga-miR-1738	−0.783	−1.721
		gga-miR-1692	−0.861	−1.816
		gga-miR-6611-5p	−1.016	−2.022
	TOC LPS 3 h (0 miRNAs)	NONE	N/A	N/A
	TOC polyI:C 3 h (0 miRNAs)	NONE	N/A	N/A
TOC 18 h	TOC AIV 18 h (2 miRNAs)	gga-miR-12234-5p	−0.61	−1.526
		gga-miR-1793	−0.629	−1.546
	TOC LPS 18 h (15 miRNAs)	gga-miR-6602-5p	−0.583	−1.498
		gga-miR-1809	−0.617	−1.533
		gga-miR-6632-5p	−0.618	−1.535
		gga-miR-6569-5p	−0.654	−1.574
		gga-miR-12266-5p	−0.666	−1.587
		gga-miR-1783	−0.682	−1.604
		gga-miR-7457-5p	−0.699	−1.624
		gga-miR-124a-5p	−0.713	−1.640
		gga-miR-2184a-3p	−0.716	−1.643
		gga-miR-12228-3p	−0.782	−1.719
		gga-miR-1761	−0.789	−1.728
		gga-miR-1795	−0.853	−1.807
		gga-miR-12234-5p	−0.915	−1.886
		gga-miR-1701	−0.92	−1.893
		gga-miR-12239-3p	−1.002	−2.003
	TOC polyI:C 18 h (40 miRNAs)	gga-miR-1735	−0.582	−1.497
		gga-miR-1551-5p	−0.606	−1.522
		gga-miR-12228-5p	−0.618	−1.535
		gga-miR-6706-5p	−0.626	−1.543
		gga-miR-7470-5p	−0.628	−1.545
		gga-miR-1692	−0.633	−1.551
		gga-miR-12248-3p	−0.663	−1.584
		gga-miR-449a	−0.68	−1.602
		gga-miR-1561	−0.712	−1.638
		gga-miR-1656	−0.713	−1.639
		gga-miR-6569-5p	−0.729	−1.658
		gga-miR-1465	−0.741	−1.672
		gga-let-7g-5p	−0.783	−1.721
		gga-miR-7480-5p	−0.803	−1.745
		gga-miR-6563-5p	−0.835	−1.784
		gga-let-7f-5p	−0.875	−1.834
		gga-miR-196-5p	−0.877	−1.836
		gga-miR-12246-5p	−0.888	−1.851
		gga-miR-1751-5p	−0.913	−1.883
		gga-miR-1671	−0.964	−1.951
		gga-miR-1593	−0.98	−1.973
		gga-miR-365-2-5p	−1.012	−2.017
		gga-miR-12208-5p	−1.021	−2.029
		gga-miR-499-5p	−1.082	−2.118
		gga-miR-1750	−1.085	−2.122
		gga-miR-130c-5p	−1.093	−2.133
		gga-miR-7458-5p	−1.093	−2.133
		gga-miR-1720-5p	−1.1	−2.144
		gga-miR-12228-3p	−1.138	−2.201
		gga-miR-1c	−1.146	−2.213
		gga-miR-7	−1.24	−2.362
		gga-miR-2184a-3p	−1.248	−2.375
		gga-miR-6564-5p	−1.286	−2.439
		gga-miR-6611-5p	−1.294	−2.452
		gga-miR-190a-5p	−1.367	−2.580
		gga-miR-1676-5p	−1.569	−2.967
		gga-miR-12248-5p	−1.944	−3.847
		gga-miR-6641-5p	−3.127	−8.733
		gga-miR-1784b-3p	−3.128	−8.742
		gga-miR-216c	−3.426	−10.751

**Table 3 vaccines-08-00438-t003:** Up-regulated miRNAs in EVs following treatment with AIV, LPS, and polyI:C. Following differential expression filtering (FC ≥ 1.5 and FDR < 0.05), 21, five, and 14 miRNAs were found to be up-regulated in EV AIV, LPS, and polyI:C groups, respectively.

Treatment Group	MiRNA	Log_2_ Fold Change	Fold Change
EV AIV (21 miRNAs)	gga-miR-301a-5p	1.525	2.877
	gga-miR-1563	1.221	2.332
	gga-miR-122-5p	0.948	1.930
	gga-miR-15b-5p	0.93	1.905
	gga-miR-1452	0.903	1.870
	gga-miR-194	0.894	1.858
	gga-miR-1575	0.812	1.756
	gga-miR-6606-5p	0.807	1.749
	gga-miR-193a-5p	0.802	1.744
	gga-miR-12221-5p	0.766	1.701
	gga-miR-6543-5p	0.727	1.655
	gga-miR-12252-3p	0.723	1.650
	gga-miR-12253-5p	0.678	1.600
	gga-miR-92-5p	0.672	1.594
	gga-miR-1670	0.672	1.593
	gga-miR-12229-5p	0.67	1.591
	gga-miR-365-1-5p	0.669	1.590
	gga-miR-6708-5p	0.64	1.558
	gga-miR-6616-5p	0.62	1.537
	gga-miR-1637	0.618	1.534
	gga-miR-383-5p	0.605	1.521
EV LPS (5 miRNAs)	gga-miR-6697-5p	0.829	1.777
	gga-miR-3535	0.697	1.621
	gga-miR-383-5p	0.649	1.568
	gga-miR-1618-5p	0.602	1.518
	gga-miR-12272-3p	0.601	1.517
EV polyI:C (14 miRNAs)	gga-miR-12253-5p	2.822	7.070
	gga-miR-12235-5p	1.713	3.279
	gga-miR-6593-5p	1.223	2.335
	gga-miR-12290-5p	0.953	1.936
	gga-miR-12252-3p	0.865	1.821
	gga-miR-1397-5p	0.856	1.810
	gga-miR-1777	0.74	1.671
	gga-miR-1456-5p	0.734	1.663
	gga-miR-6606-5p	0.714	1.640
	gga-miR-7471-5p	0.677	1.599
	gga-miR-1608	0.651	1.570
	gga-miR-1670	0.607	1.524
	gga-miR-12295-5p	0.607	1.523
	gga-miR-1649-5p	0.598	1.514

**Table 4 vaccines-08-00438-t004:** Down-regulated miRNAs in EVs following treatment with AIV, LPS, and polyI:C. Following differential expression filtering (FC ≥ 1.5 and FDR < 0.05), 57, 17, and 90 miRNAs were found to be down-regulated in EV AIV, LPS, and polyI:C groups, respectively.

Treatment Group	MiRNA	Log_2_ Fold Change	Fold Change
EV AIV (57 miRNAs)	gga-miR-3532-5p	−0.612	−1.528
	gga-miR-1722-5p	−0.619	−1.536
	gga-miR-1677-5p	−0.631	−1.549
	gga-miR-1661	−0.634	−1.552
	gga-miR-1306-5p	−0.636	−1.554
	gga-miR-2184b-5p	−0.65	−1.569
	gga-miR-107-5p	−0.661	−1.581
	gga-miR-6516-5p	−0.667	−1.588
	gga-miR-1724	−0.679	−1.601
	gga-miR-449b-5p	−0.687	−1.610
	gga-miR-1651-5p	−0.691	−1.614
	gga-miR-301b-5p	−0.696	−1.620
	gga-miR-128-1-5p	−0.7	−1.624
	gga-miR-6671-5p	−0.716	−1.643
	gga-miR-1715-5p	−0.719	−1.646
	gga-miR-365b-5p	−0.72	−1.648
	gga-miR-6675-5p	−0.729	−1.658
	gga-miR-6590-5p	−0.73	−1.659
	gga-miR-1648-5p	−0.737	−1.666
	gga-miR-726-3p	−0.746	−1.677
	gga-miR-1553-5p	−0.754	−1.687
	gga-miR-1626-5p	−0.768	−1.703
	gga-miR-1727	−0.773	−1.709
	gga-miR-6550-5p	−0.774	−1.710
	gga-miR-1664-5p	−0.784	−1.722
	gga-miR-12254-5p	−0.813	−1.757
	gga-miR-12269-3p	−0.823	−1.769
	gga-miR-1784-5p	−0.852	−1.805
	gga-miR-6639-5p	−0.865	−1.821
	gga-miR-218-5p	−0.877	−1.836
	gga-miR-12247-3p	−0.912	−1.882
	gga-miR-6665-5p	−0.913	−1.883
	gga-miR-6604-5p	−0.914	−1.885
	gga-miR-1632-5p	−1.015	−2.022
	gga-miR-1710	−1.016	−2.022
	gga-miR-1730-5p	−1.018	−2.025
	gga-miR-3536	−1.047	−2.066
	gga-miR-12244-5p	−1.078	−2.112
	gga-miR-1784b-5p	−1.089	−2.128
	gga-miR-1815	−1.096	−2.137
	gga-miR-1464	−1.096	−2.137
	gga-miR-1801	−1.1	−2.143
	gga-miR-216a	−1.101	−2.145
	gga-miR-6684-5p	−1.112	−2.161
	gga-miR-1605	−1.114	−2.165
	gga-miR-6596-5p	−1.122	−2.177
	gga-miR-12284-3p	−1.138	−2.201
	gga-miR-142-5p	−1.141	−2.205
	gga-miR-7482-5p	−1.142	−2.206
	gga-miR-7454-3p	−1.153	−2.224
	gga-miR-7464-3p	−1.256	−2.388
	gga-miR-7456-5p	−1.262	−2.398
	gga-miR-3528	−1.295	−2.454
	gga-miR-122b-3p	−1.437	−2.708
	gga-miR-210a-5p	−1.716	−3.285
	gga-miR-12260-5p	−1.791	−3.460
	gga-miR-205b	−1.865	−3.643
EV LPS (17 miRNAs)	gga-miR-1464	−0.597	−1.512
	gga-miR-132b-5p	−0.638	−1.556
	gga-miR-1584	−0.674	−1.595
	gga-miR-211	−0.689	−1.612
	gga-miR-6665-5p	−0.716	−1.643
	gga-miR-1727	−0.743	−1.674
	gga-miR-12223-3p	−0.762	−1.696
	gga-miR-1597-5p	−0.783	−1.721
	gga-miR-7482-5p	−0.79	−1.729
	gga-miR-107-5p	−0.797	−1.737
	gga-miR-449b-5p	−0.835	−1.783
	gga-miR-6557-5p	−0.869	−1.826
	gga-miR-12273-5p	−0.895	−1.859
	gga-miR-210a-5p	−1.276	−2.421
	gga-miR-12284-3p	−1.28	−2.428
	gga-miR-205b	−1.383	−2.608
	gga-miR-1784b-5p	−2.111	−4.318
EV polyI:C (90 miRNAs)	gga-miR-1465	−0.584	−1.499
	gga-miR-6582-5p	−0.589	−1.504
	gga-miR-1553-5p	−0.597	−1.512
	gga-miR-1795	−0.6	−1.516
	gga-miR-490-5p	−0.603	−1.519
	gga-miR-6707-5p	−0.604	−1.520
	gga-miR-1755	−0.627	−1.545
	gga-miR-210b-5p	−0.629	−1.546
	gga-miR-302b-5p	−0.63	−1.547
	gga-miR-211	−0.632	−1.550
	gga-miR-12274-5p	−0.644	−1.562
	gga-miR-1730-5p	−0.644	−1.562
	gga-miR-1626-5p	−0.65	−1.569
	gga-miR-23b-5p	−0.653	−1.572
	gga-miR-1667-5p	−0.657	−1.577
	gga-miR-1727	−0.671	−1.592
	gga-miR-132b-5p	−0.677	−1.598
	gga-miR-1802	−0.677	−1.598
	gga-miR-12274-3p	−0.678	−1.600
	gga-miR-1805-5p	−0.719	−1.646
	gga-miR-449a	−0.726	−1.654
	gga-miR-1597-5p	−0.729	−1.658
	gga-miR-204	−0.734	−1.663
	gga-miR-726-5p	−0.74	−1.671
	gga-miR-7444-5p	−0.741	−1.672
	gga-miR-6679-5p	−0.762	−1.696
	gga-miR-449b-5p	−0.763	−1.697
	gga-miR-6596-5p	−0.78	−1.717
	gga-miR-6567-5p	−0.78	−1.718
	gga-miR-7451-5p	−0.783	−1.720
	gga-miR-12266-5p	−0.783	−1.721
	gga-miR-212-5p	−0.789	−1.727
	gga-miR-1462-5p	−0.789	−1.728
	gga-miR-1814	−0.805	−1.747
	gga-miR-6550-5p	−0.81	−1.753
	gga-miR-6516-5p	−0.821	−1.767
	gga-miR-1690-5p	−0.83	−1.777
	gga-miR-1663-5p	−0.837	−1.787
	gga-miR-365b-5p	−0.84	−1.789
	gga-miR-1598	−0.842	−1.792
	gga-miR-301b-5p	−0.851	−1.804
	gga-miR-6598-5p	−0.882	−1.843
	gga-miR-12247-3p	−0.883	−1.844
	gga-miR-1306-5p	−0.888	−1.851
	gga-miR-6665-5p	−0.892	−1.855
	gga-miR-6604-5p	−0.899	−1.865
	gga-miR-6559-5p	−0.91	−1.879
	gga-miR-218-5p	−0.911	−1.880
	gga-miR-1715-5p	−0.919	−1.890
	gga-miR-6669-5p	−0.919	−1.891
	gga-miR-6671-5p	−0.923	−1.896
	gga-miR-12223-3p	−0.931	−1.907
	gga-miR-6566-5p	−0.932	−1.907
	gga-miR-3536	−0.964	−1.951
	gga-miR-216b	−0.966	−1.954
	gga-miR-1632-5p	−1	−2.000
	gga-miR-3532-5p	−1.006	−2.008
	gga-miR-2184b-5p	−1.009	−2.012
	gga-miR-1658-5p	−1.027	−2.037
	gga-miR-216a	−1.056	−2.080
	gga-miR-7482-5p	−1.066	−2.093
	gga-miR-1638	−1.072	−2.102
	gga-miR-449d-5p	−1.076	−2.108
	gga-miR-6639-5p	−1.079	−2.112
	gga-miR-12269-3p	−1.081	−2.116
	gga-miR-1722-5p	−1.095	−2.136
	gga-miR-7456-5p	−1.132	−2.191
	gga-miR-6675-5p	−1.14	−2.204
	gga-miR-1605	−1.155	−2.227
	gga-miR-12209-3p	−1.169	−2.248
	gga-miR-12279-3p	−1.176	−2.260
	gga-miR-12260-5p	−1.199	−2.295
	gga-miR-7479-5p	−1.207	−2.309
	gga-miR-1464	−1.214	−2.320
	gga-miR-12219-3p	−1.23	−2.346
	gga-miR-12284-3p	−1.234	−2.352
	gga-miR-1775-5p	−1.259	−2.393
	gga-miR-1573	−1.273	−2.417
	gga-miR-219a	−1.281	−2.429
	gga-miR-7473-5p	−1.308	−2.477
	gga-miR-12254-5p	−1.406	−2.651
	gga-miR-142-5p	−1.413	−2.663
	gga-miR-1677-5p	−1.418	−2.673
	gga-miR-30b-5p	−1.448	−2.728
	gga-miR-210a-5p	−1.47	−2.770
	gga-miR-19a-5p	−1.504	−2.837
	gga-miR-7454-3p	−1.571	−2.970
	gga-miR-1784b-5p	−1.654	−3.148
	gga-miR-107-5p	−1.689	−3.224
	gga-miR-205b	−2.287	−4.879

**Table 5 vaccines-08-00438-t005:** Pathway analysis summary. Host target genes for DE miRNAs were predicted using the miRDB human and mouse databases. Pathway analysis was then performed using the gene set characterization pipeline by KnowEnG. Only hits with scores greater than or equal to 95 (miRDB score ≥ 95) were considered.

Treatment Group	Total DE			Up-Regulated			Down-Regulated		
	miRNAs	Target Genes	Target Pathways	miRNAs	Target Genes	Target Pathways	miRNAs	Target Genes	Target Pathways
TOC 3 h AIV	15	105	3	11	94	1	4	11	2
TOC 3 h LPS	3	34	4	3	34	4	0	N/A	N/A
TOC 3 h polyI:C	2	151	14	2	151	14	0	N/A	N/A
TOC 18 h AIV	5	25	17	3	17	2	2	8	15
TOC 18 h LPS	47	578	14	32	447	10	15	112	4
TOC 18 h polyI:C	42	384	17	2	146	13	40	233	5
EV AIV	78	487	16	21	119	9	57	346	9
EV LPS	22	147	11	5	16	3	17	131	8
EV polyI:C	104	817	23	14	167	17	90	628	9

**Table 6 vaccines-08-00438-t006:** Prediction of pathways targeted by up-regulated cellular DE miRNAs following treatment with AIV, LPS, and polyI:C. Pathways predicted for up-regulated cellular DE miRNAs collected 3 h and 18 h post-stimulation following treatment with AIV, LPS, and polyI:C using the miRDB human and mouse database.

Treatment Group		Pathway	miRNA(s)
TOC 3 h	TOC 3 h AIV	mRNA processing	gga-let-7c-5p, gga-let-7l-5p, gga-miR-129-5p
	TOC 3 h LPS	BDNF signaling pathway	gga-miR-6704-5p
		Focal Adhesion-PI3K-Akt-mTOR-signaling pathway	gga-miR-6704-5p
		Splicing factor NOVA regulated synaptic proteins	gga-miR-1608
		Synaptic vesicle pathway	gga-miR-6704-5p
	TOC 3 h polyI:C	Imatinib resistance in chronic myeloid leukemia	gga-miR-12235-5p
		PluriNetWork	gga-miR-12235-5p
		Calcium regulation in the cardiac cell	gga-miR-12235-5p
		SIDS susceptibility pathways	gga-miR-12235-5p
		Gastric cancer network 1	gga-miR-12235-5p
		Stabilization and expansion of the E-cadherin adherens junction	gga-miR-12235-5p
		Fanconi anemia pathway	gga-miR-12235-5p
		Calcium regulation in the cardiac cell	gga-miR-12235-5p
		Focal adhesion-PI3K-Akt-mTOR-signaling pathway	gga-miR-12235-5p
		Glial cell differentiation	gga-miR-12235-5p
		Validated nuclear estrogen receptor alpha network	gga-miR-12235-5p
		Visual signal transduction: cones	gga-miR-12235-5p
		Integrin-mediated cell adhesion	gga-miR-12235-5p
		Synaptic vesicle pathway	gga-miR-12235-5p
TOC 18 h	TOC 18 h AIV	Regulation of Toll-like receptor signaling pathway	gga-miR-12244-5p
		Notch signaling pathway	gga-miR-12244-5p
	TOC 18 h LPS	Adipogenesis	gga-miR-15b-5p, gga-miR-15c-5p, gga-miR-16-5p, gga-miR-16c-5p, gga-miR-2184a-5p
		EGF/EGFR signaling pathway	gga-miR-145-5p
		Gastric cancer network 1	gga-miR-181a-5p, gga-miR-181b-5p
		Insulin signaling	gga-miR-15b-5p, gga-miR-15c-5p, gga-miR-1563, gga-miR-16-5p, gga-miR-16c-5p
		Regulation of nuclear beta catenin signaling and target gene transcription	gga-miR-145-5p
		Regulation of RAC1 activity	gga-miR-205a
		Senescence and autophagy in cancer	gga-miR-17-5p
		TarBasePathway	gga-miR-30a-5p, gga-miR-30b-5p, gga-miR-30c-5p, gga-miR-30d, gga-miR-455-5p
		TGF-beta signaling pathway	gga-miR-15b-5p, gga-miR-15c-5p, gga-miR-16-5p, gga-miR-16c-5p
		XPodNet - protein-protein interactions in the podocyte expanded by STRING	gga-miR-145-5p, gga-miR-1563, gga-miR-17-5p, gga-miR-205a, gga-miR-30a-5p, gga-miR-30b-5p, gga-miR-30c-5p, gga-miR-30d
	TOC 18 h polyI:C	Imatinib resistance in chronic myeloid leukemia	gga-miR-12235-5p
		PluriNetWork	gga-miR-12235-5p
		Calcium regulation in the cardiac cell	gga-miR-12235-5p
		SIDS susceptibility pathways	gga-miR-12235-5p
		Gastric cancer network 1	gga-miR-12235-5p
		Stabilization and expansion of the E-cadherin adherens junction	gga-miR-12235-5p
		Fanconi anemia pathway	gga-miR-12235-5p
		Calcium regulation in the cardiac cell	gga-miR-12235-5p
		Focal adhesion-PI3K-Akt-mTOR-signaling pathway	gga-miR-12235-5p
		Glial cell differentiation	gga-miR-12235-5p
		Validated nuclear estrogen receptor alpha network	gga-miR-12235-5p
		Visual signal transduction: cones	gga-miR-12235-5p
		Synaptic vesicle pathway	gga-miR-12235-5p

**Table 7 vaccines-08-00438-t007:** Prediction of pathways targeted by down-regulated cellular DE miRNAs following treatment with AIV, LPS and polyI:C. Pathways predicted for down-regulated cellular DE miRNAs collected 3 h and 18 h post-stimulation following treatment with AIV, LPS, and polyI:C using the miRDB human and mouse database.

Treatment Group		Pathway	miRNA(s)
TOC 3 h	TOC 3 h AIV	Ectoderm differentiation	gga-miR-6611-5p
		Validated targets of C-MYC transcriptional activation	gga-miR-383-5p
	TOC 3 h LPS	N/A	N/A
	TOC 3 h polyI:C	N/A	N/A
TOC 18 h	TOC 18 h AIV	Insulin Signaling	gga-miR-1793
		EGF/EGFR signaling pathway	gga-miR-1793
		ErbB1 downstream signaling	gga-miR-1793
		EGFR1 signaling pathway	gga-miR-1793
		TNF-alpha NF-kB signaling pathway	gga-miR-1793
		p38 MAPK signaling pathway	gga-miR-1793
		MAPK signaling pathway	gga-miR-1793
		Trk receptor signaling mediated by the MAPK pathway	gga-miR-1793
		Signaling mediated by p38-alpha and p38-beta	gga-miR-1793
		Serotonin receptor 4/6/7 and NR3C signaling	gga-miR-1793
		Structural pathway of interleukin 1 (IL-1)	gga-miR-1793
		LPA4-mediated signaling events	gga-miR-1793
		Interferon type I signaling pathways	gga-miR-1793
		Bladder cancer	gga-miR-1793
		BDNF signaling pathway	gga-miR-1793
	TOC 18 h LPS	Circadian rhythm related genes	gga-miR-12239-3p, gga-miR-124a-5p
		p53 signaling	gga-miR-124a-5p, gga-miR-1783
		PluriNetWork	gga-miR-124a-5p
		TNF-alpha NF-kB signaling pathway	gga-miR-7457-5p
	TOC 18 h polyI:C	Ectoderm differentiation	gga-miR-1c, gga-miR-6611-5p, gga-miR-6706-5p
		Integrated breast cancer pathway	gga-miR-1c, gga-miR-130c-5p
		mRNA processing	gga-miR-12248-5p, gga-miR-1465, gga-miR-7, gga-let-7f-5p, gga-let-7g-5p
		PluriNetWork	gga-miR-1551-5p, gga-miR-449a
		TNF-alpha NF-kB signaling pathway	gga-miR-6641-5p, gga-miR-7480-5p

**Table 8 vaccines-08-00438-t008:** Prediction of pathways targeted by up-regulated EV DE miRNAs following treatment with AIV, LPS and polyI:C. Pathways predicted for up-regulated EV DE miRNAs collected 3 h and 18 h post-stimulation following treatment with AIV, LPS, and polyI:C using the miRDB human and mouse database.

Treatment Group	Pathway	miRNA(s)
EV AIV	FOXA1 transcription factor network	gga-miR-15b-5p
	GPCRs, Class A Rhodopsin-like	gga-miR-6616-5p
	Insulin signaling	gga-miR-1563, gga-miR-15b-5p
	mir-124 predicted interactions with cell cycle and differentiation	gga-miR-92-5p
	mRNA processing	gga-miR-15b-5p, gga-miR-1452, gga-miR-6543-5p
	p73 transcription factor network	gga-miR-194
	TGF-beta signaling pathway	gga-miR-15b-5p
	Validated targets of C-MYC transcriptional activation	gga-miR-383-5p
	Validated targets of C-MYC transcriptional repression	gga-miR-6708-5p
EV LPS	Matrix Metalloproteinases	gga-miR-12272-3p
	mRNA Processing	gga-miR-12272-3p
	Validated targets of C-MYC transcriptional activation	gga-miR-383-5p
EV polyI:C	Calcium regulation in the cardiac cell	gga-miR-12235-5p
	Fanconi anemia pathway	gga-miR-12235-5p
	Focal adhesion-PI3K-Akt-mTOR-signaling pathway	gga-miR-12235-5p
	Gastric cancer network 1	gga-miR-12235-5p
	Glial cell differentiation	gga-miR-12235-5p
	Imatinib resistance in chronic myeloid leukemia	gga-miR-12235-5p
	PluriNetWork	gga-miR-12235-5p
	SIDS Susceptibility pathways	gga-miR-12235-5p
	Splicing factor NOVA regulated synaptic proteins	gga-miR-1608
	Stabilization and expansion of the E-cadherin adherens junction	gga-miR-12235-5p
	Synaptic vesicle pathway	gga-miR-12235-5p
	Validated nuclear estrogen receptor alpha network	gga-miR-12235-5p
	Visual signal transduction: cones	gga-miR-12235-5p
	XPodNet - protein-protein interactions in the podocyte expanded by STRING	gga-miR-1456-5p

**Table 9 vaccines-08-00438-t009:** Prediction of pathways targeted by down-regulated EV DE miRNAs following treatment with AIV, LPS, and polyI:C. Pathways predicted for down-regulated EV DE miRNAs collected 3 h and 18 h post-stimulation following treatment with AIV, LPS, and polyI:C while using the miRDB human and mouse database.

Treatment Group	Pathway	miRNA(s)
EV AIV	Caspase cascade in apoptosis	gga-miR-1784b-5p, gga-miR-3536
	Direct p53 effectors	gga-miR-12247-3p, gga-miR-12284-3p, gga-miR-205b, gga-miR-142-5p
	Imatinib resistance in chronic myeloid leukemia	gga-miR-1724, gga-miR-6639-5p
	Regulation of RAC1 activity	gga-miR-205b, gga-miR-449b-5p
	Splicing factor NOVA regulated synaptic proteins	gga-miR-7456-5p
	Synaptic vesicle pathway	gga-miR-1632-5p, gga-miR-3532-5p, gga-miR-6639-5p
	TGF-beta signaling pathway	gga-miR-1632-5p, gga-miR-142-5p, gga-miR-1727
	Validated targets of C-MYC transcriptional repression	gga-miR-12247-3p, gga-miR-1626-5p
	XPodNet - protein-protein interactions in the podocyte expanded by STRING	gga-miR-301b-5p, gga-miR-1632-5p, gga-miR-6639-5p, gga-miR-205b, gga-miR-218-5p, gga-miR-107-5p
EV LPS	Caspase Cascade in apoptosis	gga-miR-1784b-5p
	Direct p53 effectors	gga-miR-12284-3p, gga-miR-205b
	Globo sphingolipid metabolism	gga-miR-1597-5p, gga-miR-211
	PluriNetWork	gga-miR-449b-5p
	PodNet: protein-protein interactions in the podocyte	gga-miR-107-5p, gga-miR-205b
	Regulation of RAC1 activity	gga-miR-205b, gga-miR-449b-5p
	Stabilization and expansion of the E-cadherin adherens junction	gga-miR-211
	XPodNet - protein-protein interactions in the podocyte expanded by STRING	gga-miR-107-5p, gga-miR-205b, gga-miR-211
EV polyI:C	BMP receptor signaling	gga-miR-1677-5p, gga-miR-490-5p, gga-miR-7454-3p
	Circadian rhythm related genes	gga-miR-1632-5p, gga-miR-1755, gga-miR-218-5p, gga-miR-365b-5p, gga-miR-7482-5p
	Direct p53 effectors	gga-miR-12247-3p, gga-miR-12284-3p, gga-miR-142-5p, gga-miR-205b, gga-miR-219a
	mRNA processing	gga-miR-1465, gga-miR-1638, gga-miR-1663-5p, gga-miR-205b, gga-miR-6598-5p, gga-miR-726-5p
	PodNet: protein-protein interactions in the podocyte	gga-miR-107-5p, gga-miR-1632-5p, gga-miR-1658-5p, gga-miR-205b, gga-miR-219a, gga-miR-301b-5p
	Regulation of RAC1 activity	gga-miR-205b, gga-miR-449a, gga-miR-449b-5p, gga-miR-449d-5p, gga-miR-7451-5p
	Splicing factor NOVA regulated synaptic proteins	gga-miR-30b-5p, gga-miR-302b-5p, gga-miR-7456-5p
	Synaptic vesicle pathway	gga-miR-132b-5p, gga-miR-1632-5p, gga-miR-3532-5p, gga-miR-6639-5p, gga-miR-6669-5p
	XPodNet – protein-protein interactions in the podocyte expanded by STRING	Gga-miR-107-5p, gga-miR-1632-5p, gga-miR-1658-5p, gga-miR-204, gga-miR-205b, gga-miR-211, gga-miR-218-5p, gga-miR-219a, gga-miR-30b-5p, gga-miR-301b-5p, gga-miR-6598-5p, gga-miR-6639-5p, gga-miR-6669-5p

**Table 10 vaccines-08-00438-t010:** MiRNAs targeting the viral genome. Viral target sites for DE miRNAs were predicted using the miRanda and RNAhybrid algorithms. MiRNAs targeting specific segments are indicated, along with targets positions, miRanda scores, free energy and expression (up- or down-regulation) within the treatment groups.

Segment	Protein(s)	miRNA	Position	miRanda score	miRanda Free Energy (kcal/mol)	Expression
Segment 1	PB2	gga-miR-122b-3p	1207–1227	170	−17.54	Downregulated in EV AIV
		gga-miR-146a-5p	1896–1917	161	−17.51	Upregulated in TOC LPS 18 h
		gga-miR-146b-5p	1896–1917	161	−17.87	Upregulated in TOC LPS 18 h
		gga-miR-1720-5p	1249–1268	177	−32.16	Downregulated in TOC polyI:C 18 h
		gga-miR-6671-5p	578–601	161	−22.53	Downregulated in EV AIV & EV polyI:C
Segment 2	PB1, PB1-F2	gga-miR-107-5p	774–796	163	−20.95	Downregulated in EV AIV, EV LPS & EV polyI:C
		gga-miR-12223-3p	1328–1349	160	−27.23	Downregulated in EV LPS & EV polyI:C
		gga-miR-129-5p	2264–2284	179	−24.92	Upregulated in TOC AIV 3h
		gga-miR-132b-5p	91–111	160	−20.56	Downregulated in EV LPS & EV polyI:C
		gga-miR-1661	1822–1844	163	−26.29	Downregulated in EV AIV
		gga-miR-6641-5p	1729–1750	176	−16.88	Downregulated in TOC polyI:C 18 h
Segment 3	PA	gga-miR-1573	1027–1047	166	−19.85	Downregulated in EV polyI:C
		gga-miR-1663-5p	266–285	162	−26.14	Downregulated in EV polyI:C
		gga-miR-1715-5p	1439–1463	161	−24.17	Downregulated in EV AIV & EV polyI:C
		gga-miR-6665-5p	2005–2025	162	−20.58	Downregulated in EV AIV, EV LPS & EV polyI:C
		gga-miR-7454-3p	944–968	160	−19.88	Downregulated in EV AIV & EV polyI:C
Segment 4	HA	gga-miR-1593	500–520	161	−21.16	Downregulated in TOC polyI:C 18 h
		gga-miR-1605	55–76	168	−20.89	Downregulated in EV AIV & EV polyI:C
		gga-miR-1671	761–783	161	−24.1	Downregulated in TOC polyI:C 18 h
Segment 5	NP	gga-miR-12269-3p	67–88	160	−22.67	Downregulated in EV AIV & EV polyI:C
		gga-miR-145-5p	291–313	167	−22.82	Upregulated in TOC LPS 18 h
		gga-miR-1784b-5p	1163–1186	163	−21.01	Downregulated in EV AIV, EV LPS & EV polyI:C
		gga-miR-6679-5p	786–809	178	−20.29	Downregulated in EV polyI:C
Segment 6	NA	gga-miR-1783	455–477	163	−20.22	Downregulated in TOC LPS 18 h
		gga-miR-218-5p	535–553	160	−23.78	Downregulated in EV AIV & EV polyI:C
Segment 7	M1, M2	gga-miR-1710	569–590	164	−19.87	Downregulated in EV AIV
		gga-miR-1784b-5p	565–588	161	−18.81	Downregulated in EV AIV, EV LPS & EV polyI:C
Segment 8	NS1, NEP	NONE	N/A	N/A	N/A	N/A

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
