# Peer review of "Distinct miRNA Profile of Cellular and Extracellular Vesicles Released from Chicken Tracheal Cells Following Avian Influenza Virus Infection"

_vaccines, 2020, doi:10.3390/vaccines8030438_

Round 1
Reviewer 1 Report
This manuscript confirms that cells in tracheal organ culture (TOC) release extracellular vesicles and that the miRNA population changes in the cells and in the EVs according to the stimulation they receive. These is not surprising at all. Three types of stimuli were used here: AIV infection, Poly I:C treatment and LPS treatment and an enormous amount of data was generated and is presented to the reader without sufficient analysis. The few analysis performed are not well explained (FIG.5).
The authors should consider to move tables to the supplementary material section and perform more analyses using Venn diagrams to make easier for the reader to visualize the information.
It is impossible to read the text in the figures. The quality of the graphs is unacceptable.
Table 3, is currently labeled as Table 1.
Summarizing, the amount of work is impressive but the paper is very difficult to read, in part because the data is insufficiently analyzed, improperly represented (no Venn diagrams or bar graphs that make more intuitive the results). The conclusions are based on what the authors decided to extracted from the data but I am not sure that is a complete representation of what happened in reality. The conclusions do not indicate that after this work the understanding in the field of EV or innate immunity gained any progress.
Author Response
Reviewer 1
[Q]: This manuscript confirms that cells in tracheal organ culture (TOC) release extracellular vesicles and that the miRNA population changes in the cells and in the EVs according to the stimulation they receive. These is not surprising at all. Three types of stimuli were used here: AIV infection, Poly I:C treatment and LPS treatment and an enormous amount of data was generated and is presented to the reader without sufficient analysis. The few analysis performed are not well explained (FIG.5).
Response:
We thank the reviewer for comments which help us to improve the manuscript.
In this study, we hypothesized that tracheal cells produce EVs, and their contents are affected by the type of stimuli. As the reviewer mentioned, it is an assumption and may not be surprising. However, there was not any scientific paper in chickens to demonstrate the release of Evs from tracheal cells. In addition, for the first time, we compared the cellular and EV miRNAs following AIV infection or TLR ligand stimulation, which demonstrate the novelty of this work. By comparing the profile of miRNAs in EVs between different groups or between cellular and EV miRNAs, the results of our study showed that the packaging system of EVs is affected by the type of stimuli, and it is not a random process.
Concerning the reviewer’s comment about the lack of sufficient statistical analysis, we have done statistical analysis in each step to present differentially expressed (DE) miRNAs. In the first step, we determined DE miRNAs comparing to control groups for each treatment individually considering their fold change as well as False Discovery Rate, which is the standard approach to analyze expression data analysis following next-generation sequencing. In addition, as we indicated in the manuscript, MiRNA differential expression modeling and calculation were done using R and R packages edgeR, tidyverse, magrittr, and ComplexHeatmap.
Functional annotation analysis is the next step in the routine analysis pipeline, including Gene Ontology over-representation analysis and pathway analysis. In the current study, we performed both types of functional annotation analysis on an artificial intelligence-based platform which was developed for this study. It is beyond most of the descriptional studies that only perform functional analysis on the web-based public search engines, such as miRDB or commercial software.
The reviewer could find examples of such papers from the authors of this manuscript or previously published papers by others:
1- Transcriptomic Profiles of Monocyte-Derived Macrophages in Response to Escherichia coli is Associated with the Host Genetics. Emam M, Cánovas A, Islas-Trejo AD, Fonseca PAS, Medrano JF, Mallard B. Sci Rep. 2020 Jan 14;10(1):271. doi: 10.1038/s41598-019-57089-0.
2- Identification and characterization of differentially expressed exosomal microRNAs in bovine milk infected with Staphylococcus aureus. Ma S, Tong C, Ibeagha-Awemu EM, Zhao X. BMC Genomics. 2019 Dec 5;20(1):934. doi: 10.1186/s12864-019-6338-1.
3- Integrated analysis of microRNA expression and mRNA transcriptome in lungs of avian influenza virus infected broilers. Ying Wang, Vinayak Brahmakshatriya, Blanca Lupiani, Sanjay M Reddy, Benjamin Soibam, Ashley L Benham, Preethi Gunaratne, Hsiao-ching Liu, Nares Trakooljul, Nancy Ing, Ron Okimoto, Huaijun Zhou. BMC Genomics. 2012 Jun 22;13:278. doi: 10.1186/1471-2164-13-278.
We are happy to consider any further analysis if the reviewer can specifically recommend additional analysis.
In fact, Fig.5 is a new replacement of Venn diagrams with the same principle, which only presents in-common and group-specific numbers. Neither Venn diagrams nor UpSEt diagrams require any statistical analysis between the elements presented in the graph. Although Venn diagrams are easy to understand, we chose to use UpSet diagram because of zero overlaps between some groups. The UpSet diagram is the new replacement of the Venn diagram, which is becoming more popular in the scientific community. Here are some examples:
1- miRspongeR: an R/Bioconductor package for the identification and analysis of miRNA sponge interaction networks and modules. Junpeng Zhang, Lin Liu, Taosheng Xu, Yong Xie, Chunwen Zhao, Jiuyong Li, Thuc Duy Le. BMC Bioinformatics. 2019 May 10;20(1):235. doi: 10.1186/s12859-019-2861-y.
2- Global identification of functional microRNA-mRNA interactions in Drosophila. Hans-Hermann Wessels, Svetlana Lebedeva, Antje Hirsekorn, Ricardo Wurmus, Altuna Akalin, Neelanjan Mukherjee, Uwe Ohler. Nat Commun. 2019 Apr 9;10(1):1626. doi: 10.1038/s41467-019-09586-z.
In the end, we changed Fig5 in the revised file and used Venn diagrams based on the reviewer’s comment.
[Q]: The authors should consider to move tables to the supplementary material section and perform more analyses using Venn diagrams to make easier for the reader to visualize the information.
Response:
We thank the reviewer for the comment. We do not have any preference to keep the table in the main manuscript or move them to the supplementary. If the editor recommends moving the tables, we are happy to do it. We can consult with the editorial group to present tables in a better format and keep them in the paper as we believe they offer essential information for the readers. We already have several tables in the supplementary file.
[Q]: It is impossible to read the text in the figures. The quality of the graphs is unacceptable.
Response:
We have uploaded high-quality figures. For sure, we will communicate with the journal to make sure of having high-quality figures/graphs for the publication.
[Q]: Table 3, is currently labeled as Table 1.
Response: We revised it.
[Q]: Summarizing, the amount of work is impressive, but the paper is very difficult to read, in part because the data is insufficiently analyzed, improperly represented (no Venn diagrams or bar graphs that make more intuitive the results). The conclusions are based on what the authors decided to extracted from the data but I am not sure that is a complete representation of what happened in reality. The conclusions do not indicate that after this work the understanding in the field of EV or innate immunity gained any progress.
Response:
Thanks for the comment. As the reviewer mentioned, we had to focus on the part of data that was aligned with our hypothesis and objectives (induction of antiviral responses in the context of avian influenza virus). Unfortunately, this is the downside of working with big data, which we can not focus on the entire generated data. However, we will share our row and analyzed data on Gene Expression Omnibus to make it freely available to the scientific community. It will let other research groups re-analyze the data based on their aims or objectives.
In the current study, we have identified several miRNAs which potentially target important pathways involved in antiviral responses. These miRNAs could be candidates for future functional studies. We reported miRNAs, such as miR-146a, miR-146b, miR-205a, miR205b and miR-449, which are potentially involved in antiviral responses.
In addition, the current study is the base for future studies in the context of exosomes in the chicken respiratory system. Currently, we continue this work with a focus on the immune-regulatory function of exosomes and their contents. Therefore, we believe that this manuscript has provided vital information about exosomes released from tracheal cells in chickens. In addition, analysis of exosomal contents provided information for future studies aiming to identify mechanisms involved in cell to cell communications in chickens.

Reviewer 2 Report
Although descriptive, this study may set the basis for further investigation.
Please note that the quality/resolution of the figures is very low.
Author Response
Comments and Suggestions for Authors
Although descriptive, this study may set the basis for further investigation.
Please note that the quality/resolution of the figures is very low.
Response:
We appreciate the reviewer’s comment. We have uploaded high-quality figures. For sure, we will communicate with the journal to make sure of having high-quality figures/graphs for the publication.
Reviewer 3 Report
This paper characterises differential expression of chicken tracheal miRNAs following immune cell simulation with AIV, LPS and polyI:C. The identification of common miRNAs among the different treatment groups – which could indicate a fundamental role in EV functionality – provide promising opportunities for future research. The methods are robust, the results clearly presented and the discussion comprehensive. Accordingly, my comments are minor. To facilitate reproducibility, could the raw data be uploaded to a public repository? Furthermore, given pre-processing is critical to analyses of miRNA reads, what parameters were used for Trimmomatic?
Finally, some minor grammatical points:
Line 180. Remove ‘and’ from ‘to confirm and biomarkers’
Line 206. Erroneous capitalisation of ‘a’ in ‘Briefly, A…’
Author Response
This paper characterises differential expression of chicken tracheal miRNAs following immune cell simulation with AIV, LPS and polyI:C. The identification of common miRNAs among the different treatment groups – which could indicate a fundamental role in EV functionality – provide promising opportunities for future research. The methods are robust, the results clearly presented and the discussion comprehensive. Accordingly, my comments are minor. To facilitate reproducibility, could the raw data be uploaded to a public repository? Furthermore, given pre-processing is critical to analyses of miRNA reads, what parameters were used for Trimmomatic?
Response:
Absolutely, we will share our row and analyzed data on Gene Expression Omnibus to make it freely available to the scientific community.
This part of the analysis was performed by the Canadian center for computational genomics. Here are the parameters which were set for the analysis:
java -jar trimmomatic.jar SE <INPUT_FASTQ> <OUTPUT_FASTQ> ILLUMINACLIP:adapters.fa:2:30:10 SLIDINGWINDOW:4:15 MINLEN:18
The most important deviations from the defaults are:
- the “adapters.fa” file included the adapter sequence “AGATCGGAAGAGCACACGTCTGAACTCCAGTCA”
- The minimum length read is reduced to 18, as the reads are already only 51 bp.
Graph (The sequencing quality score has been uploaded in the attached file)
As it is presented in the graph, the majority of reads after QC had a quality of greater than 30.
[Q] Finally, some minor grammatical points:
Line 180. Remove ‘and’ from ‘to confirm and biomarkers’
Line 206. Erroneous capitalisation of ‘a’ in ‘Briefly, A…’
Response: We have revised the manuscript.

Round 2
Reviewer 1 Report
The authors have answer the comments satisfactorily and I consider the manuscript has been improved respect its previous version.